# Bu-M-P-ing Iron: How BMP Signaling Regulates Muscle Growth and Regeneration

**DOI:** 10.3390/jdb8010004

**Published:** 2020-02-11

**Authors:** Matthew J Borok, Despoina Mademtzoglou, Frederic Relaix

**Affiliations:** 1Inserm, IMRB U955-E10, 94010 Créteil, France; matthew.borok@inserm.fr (M.J.B.); despoina.mademtzoglou@inserm.fr (D.M.); 2Faculté de santé, Université Paris Est, 94000 Creteil, France; 3Ecole Nationale Veterinaire d’Alfort, 94700 Maison Alfort, France; 4Etablissement Français du Sang, 94017 Créteil, France; 5APHP, Hopitaux Universitaires Henri Mondor, DHU Pepsy & Centre de Référence des Maladies Neuromusculaires GNMH, 94000 Créteil, France

**Keywords:** development, regeneration, TGFβ, stem cells, satellite cells

## Abstract

The bone morphogenetic protein (BMP) pathway is best known for its role in promoting bone formation, however it has been shown to play important roles in both development and regeneration of many different tissues. Recent work has shown that the BMP proteins have a number of functions in skeletal muscle, from embryonic to postnatal development. Furthermore, complementary studies have recently demonstrated that specific components of the pathway are required for efficient muscle regeneration.

## 1. Introduction

As its name implies, the bone morphogenetic protein (BMP) signaling pathway was first identified as a key regulator of bone formation. Urist and colleagues isolated proteins from rabbit bone and then used them to induce bone formation in vitro or in vivo in the rat [1]. Later work showed that these proteins were highly conserved during evolution and most species had a number of paralogues [2,3,4]. Mapping of the mouse short ear gene, essential for skeletal patterning and bone repair, to the Bmp5 locus, was the first demonstration of the importance of this pathway in mice [5]. In contrast to the relatively mild phenotype of Bmp5 deletions in short-eared mice, deletion of either Bmp4 or Bmp2 and their receptors Bmpr1a and Bmpr2 caused early embryonic lethality, with severe defects in mesoderm formation, extraembryonic tissues and cardiac development [6,7,8,9] (Table A1). In addition to demonstrating that certain BMP genes are essential for development, this work implicated this signaling pathway to play important roles in multiple organ systems.

## 2. Signaling Mechanism

After decades of biochemical work, many components of the BMP signaling pathway and their mode of action have been described. Signaling activation requires two different receptors, first discovered by experiments with a human cell line [10]. Today the pathway consists of at least 20 different ligands, four type 1 receptors, and three type 2 receptors, varying combinations of which give distinct responses [11]. After binding of the BMP ligand, the receptors phosphorylate Smad cofactors, allowing an interaction with Smad4 and translocation of the Smad complex into the nucleus (Figure 1) [12,13,14]. The BMP-specific cofactors are Smad1, Smad5, and Smad9. Surprisingly, while single knockouts for Smad1 and Smad5 are embryonic lethal, Smad9 is dispensable for development [15,16,17,18,19]. The interaction between Smad1/5/8 and Smad4 is BMP-specific, while TGFβ signaling induces an interaction between Smad2/3 and Smad4 [20,21,22]. Furthermore, Smad1 and Smad4 have homologous transcriptional activation domains and their nuclear import leads to transcriptional competence [14,23]. While there is some overlap in the two pathways, the specific components, including extracellular and intracellular inhibitors, receptors, and Smads distinguish BMP from TGFβ signaling, leading to distinct outcomes and specific developmental consequences [24,25]. The regulation of skeletal muscle growth and regeneration by the TGFβ pathway falls beyond the scope of this review; readers may refer to previous works [26,27].

The most common BMP targets are Id and Msx genes. The canonical downstream target of BMP signaling is Id1, first identified in osteoblast-like cells [28]. Later work in embryonic stem cells and a number of different cell lines confirmed this target and also identified Id2, Id3, Msx1, and Msx2 as direct targets of the pathway [29]. The *Drosophila* homologue of the Smad proteins, Mad, binds to a GCCG motif in target genes [30]. When this site is present in a multimerized form, it is sufficient to bind Smad proteins and to confer BMP-specific activation in several mammalian cell lines [31]. 

Several inhibitors of the pathway have been identified. Work in *Xenopus* identified Smad6 as an inhibitor of the pathway, which functions by blocking Smad4 association with Smad1 (Figure 1) [32]. A number of secreted BMP inhibitors have also been described. *Xenopus* explant experiments demonstrated that Bmp4 and noggin have opposing effects on embryonic patterning [33]. Biochemical experiments demonstrated that noggin is able to bind to BMP ligands, rendering them incapable of binding to their receptors [34]. Related antagonists chordin and gremlin, function in the same way (Figure 1, Table A1) [35,36,37].

## 3. Role in Skeletal Muscle

Experiments with rodent myogenic cell lines were the first to demonstrate an inhibitory role for BMP signaling on muscle differentiation (Figure 2) [38,39]. The addition of BMP ligands to the myogenic L6 and C2C12 cell lines, as well as primary rat cell cultures, inhibited the formation of myotubes and the expression of the myogenic differentiation genes myogenin, MyoD, and Myf5, while it promoted a concomitant trans-differentiation into osteogenic fate [40,41,42,43,44,45,46,47,48,49,50]. Of note, Bmp2-activated muscle tissue was as efficient as autologous bone grafting in the treatment of large bone defects [51]. Furthermore, the implantation of BMP2-soaked beads in rat calf muscle resulted in osteogenic differentiation [52]. On the contrary, culture of myogenic cells in the presence of BMP inhibitors promoted myogenic differentiation (Figure 2, Table A1) [42,43,44,45,46,49,53,54]. However, the role of BMP signaling in muscle differentiation appears to be much more complex in embryonic development and adult muscle regeneration.

## 4. Regulation of Prenatal Muscle Development

Skeletal muscle is of mesodermal origin, with paraxial mesoderm contributing to limb and trunk muscles, starting by a segmentation process called somitogenesis. As somites mature, dermomyotomal cells will either form the myotome by intercalating between the dorsal dermomyotome and ventral sclerotome or delaminate and migrate from the hypaxial domain specific somites to form the muscle masses of the limbs, diaphragm, and tongue [55,56,57,58]. Stem cells of the developing trunk muscle will then migrate from the dermomyotome into the myotome [59,60,61,62]. On the molecular level, Pax proteins are known to be involved in the control of many lineages during embryogenesis, with Pax3 and Pax7 acting as key regulators in the muscle lineage. Specifically, they orchestrate progenitor survival, proliferation, migration, self-renewal, and they trigger the myogenic program. At later stages, myogenic determination and differentiation are ensured by a family of bHLH transcription factors, collectively referred to as myogenic regulatory factors (family members: Myf5, MyoD, Mrf4/Myf6, myogenin) [63]. With the core myogenic program being very well described, several groups are trying to address impact of different factors or pathways, including the BMP pathway, on this program.

The first studies of BMP in muscle development were largely concerned with the role of the pathway in axis patterning and repression of the myogenic program. BMP diverts paraxial mesoderm fate towards lateral-plate fate, while BMP loss-of-function leads to expansion of the paraxial mesoderm domain [64]. Several groups showed that exogenous BMP ligands could skew *Xenopus* and avian mesoderm towards a ventral fate, at the expense of dorsal fates, including muscle [35,65,66,67,68], while these findings were more recently confirmed by Smad1 addition to *Xenopus* dorsal mesoderm [43]. In contrast, addition of the BMP inhibitors noggin or chordin to *Xenopus* embryo explants or chick embryos was able to drastically increase transcripts of *muscle actin* and *MyoD*, respectively (Table A1) [33,35,69,70]. In keeping with a muscle-repressive role, experiments in chick embryo explants demonstrated that BMP signaling favored the differentiation of cartilage over muscle [71]. Of note, co-injection of noggin and constitutively active BMP receptor led to a failure of noggin to dorsalize *Xenopus* embryos [34]. Thus, initial studies argued for a repressor role of BMP on myogenic fate.

The idea that BMP universally repressed muscle formation was soon challenged (Table A1) by explant experiments demonstrating that Bmp4 acted as a morphogen in *Xenopus* embryos [72]. A specific range of concentrations actually promoted the expression of Myf5 and formation of muscle. Furthermore the antagonizing function of noggin led to a gradient of Bmp4 activity distinct from the expression pattern of Bmp4. In chick embryo explants, exogenous Bmp4 had no effect on expression of the broad dermomyotomal marker Pax3, but repressed genes involved in myogenic differentiation including MyoD [70]. Further work in chick embryos implicated ectodermal and mesenchymal tissue Bmp2, Bmp4, and Bmp7 in sustaining Pax3 expression in myogenic progenitors of the limb and trunk as well as in proliferation of these progenitors [57,73,74,75]. This effect was shown to be dose-dependent in the limb [73]. Ectopic Bmp4 or constitutively active Bmpr1a during fetal myogenesis increased the number of Pax7+ and differentiating (MF20+) cells, while ectopic Bmp4 and ectopic noggin at E18 respectively increased and decreased the number of satellite cells, which are the stem cells of adult skeletal muscle [76]. Cranial myogenesis also relies upon a balance of ectodermal BMP ligands and the BMP antagonists noggin and gremlin produced in the cranial neural crest and other head tissues [77], while Bmp2 was found to negatively affect myogenic differentiation in embryonic tongues [78]. Specifically, in vivo transduction of cranial paraxial mesoderm with Bmp4 leads to loss of the branchial arch musculature, while activation of skeletal myogenesis correlates with the presence of noggin/gremlin-expressing cranial neural crest cells [77]. The above studies point to an essential contribution of BMP effectors and inhibitors on proper muscle formation.

While it is clear that BMP signaling plays a key role in myogenesis, it remains ambiguous exactly which cells of the muscle express BMP ligands and antagonists, and which cells transduce BMP signals in vivo. During zebrafish muscle development, BMP was specifically absent from the slow-twitch muscle pioneers and medial fast-twitch fibers (MFFs) of the central domain of the myotome, while BMP inhibition by dorsomorphin or Bmpr1a morpholino treatment led to premature activation of the pioneer marker Engrailed and subsequent increase in the numbers of pioneers and MFFs [79,80,81]. Analysis of the BMP expression pattern in chick embryos revealed Bmp2 and Bmp4 presence in the posterior margin and apical ectodermal ridge (Bmp2) or anterior mesenchyme (Bmp4) [81], while Bmp4-expressing cells were flanking Pax3-expressing cells in the trunk [73,74]. In line with these findings, Pax7+ dermomyotomal cells in zebrafish embryos co-expressed pSmad1/5/8 and there was an increase in Pax7+ cells in response to Bmp2b overexpression [82]. On the contrary, noggin was expressed in the mesenchyme core of the developing limb bud, in close proximity to MyoD-expressing cells [73] and in the dorsomedial lip of the dermomyotome of more rostral somites [70]. Work in fetal human muscle showed that most Bmp4- and Bmpr1a-expressing cells were located in between developing fibers, while gremlin was expressed in interstitial progenitor cells and fibers [83]. Ex vivo, sorted Bmp4-expressing cells could give rise to muscle, while they were able to inhibit the differentiation of gremlin-expressing cells. Immunostaining for phosphorylated Smad1/5/8 in chick embryos showed that muscle progenitors at the tips of developing muscles adjacent to tendon were transducing BMP signals [76]. Implantation of Bmp2- or noggin-soaked beads changed the domains of Tcf4-expressing cells [81], a population that has been shown to pre-pattern limb muscles [84]. Furthermore, increasing and decreasing the levels of BMP signaling with viruses led to an expansion or reduction, respectively, of muscle progenitors and muscle size. Genetically increasing BMP signaling by knocking out noggin in mice led to lack or malformation of many muscles, decrease in thick fibers and reduction in Pax7+ cells [85], confirming previous findings on Pax7+ cells by ectopic noggin expression [76]. Analysis of noggin KO in different backgrounds showed that even though the early specification (either Pax3+ progenitor cells or MyoD+ precursor cells) was unaffected, the terminal stages of myogenesis were delayed, with mutant E14.5 embryos and newborns having less fusion and disorganized muscles, respectively [86]. *Chordino* mutant zebrafish embryos, which lack the chordino BMP inhibitor, exhibit upregulated BMP leading to a dorsally expanded Bmp4 expression domain and enlarged tail somites [87,88]. 

Three groups tried to address when BMP has a stronger impact on myogenesis, analyzing more or less advanced stages of myogenic progression during prenatal development. In an ex vivo system to develop Pax7+ myogenic progenitors from *Xenopus* embryos, it was shown that Bmp4 overexpression increased Pax7+ cells and Pax7 expression, an effect counteracted by induction of heat-shock-inducible noggin. Bmp4 treatment even induced paraxial explants to develop Pax7+ cells, although such cells are not formed normally from paraxial tissue. However, this was only observed when the treatment was done at stage 13, but not 18 or 22. The findings were confirmed in vivo, with Bmp4 and the BMP downstream target Msx1 increasing the number of Pax7+ cells, when injected in *Xenopus* embryos [89]. Transcriptional analysis of Myf5-expressing cells from embryonic and fetal mouse muscle revealed a higher expression of the BMP target genes Id2 and Smad6 in embryonic myoblasts [90]. Bmp4 addition to cultured embryonic myoblasts had no effect, while it inhibited myotube formation in fetal cultures [90]. Together these results suggested that BMP signaling supports embryonic myogenesis but represses muscle differentiation in fetal stages. In keeping with a potential role of BMP in earlier stages of the myogenic differentiation, Bmpr1a ablation in Myf5+ cells led to smaller muscles comprised of less and smaller myofibers and fat accumulation, while Bmpr1a ablation in MyoD+ cells did not affect muscle mass [91]. Overall, it appears that BMP signaling plays a more important role in early prenatal stages of myogenic differentiation.

## 5. Regulation of Early Postnatal Muscle Growth

The passage from small, proliferative prenatal myogenic populations to a large, post-mitotic adult tissue requires a critical early postnatal period with major changes in terms of hyperplasia, hypertrophy, cell-cycle status and acquisition of quiescence. In the late fetal stage, muscle progenitors are progressively embedded under the basal lamina [59,60,61,62]. At that point, they are described as muscle satellite cells, providing the pool of muscle stem cells required for postnatal growth and muscle repair after achieving quiescence at adult stage [92]. Only a limited number of studies investigated the role of BMP in the crucial early postnatal phase of muscle formation. Several components of the pathway were shown to be expressed both in whole muscle extracts and in satellite cells at different time-points from P3 to P28 in mice and rats (Table A1) [93,94]. Using the muscle fiber-specific myosin light chain 1 fast Cre, Sartori and colleagues found that deletion of Smad4 in muscle fibers led to decreased muscle size (Table A1) [95]. However, in that study it was not investigated whether smaller muscles were the result of defects in embryonic, fetal or postnatal myogenesis. Moreover, the deletion of Smad4 blocks both TGFβ and BMP signaling, making result interpretation difficult. For instance, Myf5-Cre:Smad4^f/f^ mice have small tongues prenatally, which was attributed to the loss of TGFβ signaling rather than BMP, as revealed by subsequent analysis [96]. Hence, BMP is likely to play a role in postnatal myogenesis, but the negative impact of Smad4 blockade on muscle formation should be cautiously interpreted, as Smad4 is in the interconnection of the TGFβ and BMP pathways.

Specific blockade of the BMP pathway with multiple genetic and pharmacological studies in neonatal mice helped clarify the situation [94]. Stantzou and colleagues showed that the loss of BMP signaling in satellite cells led to decreased satellite cell numbers, reduced muscle weight and fiber length. Thus BMP signaling appeared to play a similar role as in embryogenesis by maintaining proliferation of activated satellite cells. Furthermore, introduction of exogenous BMPs into postnatal muscle led to pronounced hypertrophy [97]. Upstream of BMP, miR-26a seems to target Smad1, Smad4, and the BMP target Id3 [98]. Blocking miR-26a in neonatal mice increased proliferation and decreased myogenic differentiation [98]. An unexpected role for the BMP pathway in muscle was found when deleting the receptor Bmpr1a in MyoD- and Myf5-expressing cells [91]. The epaxial muscles of mutants contained smaller myofibers and increased fat infiltration, potentially due to the role of this pathway in endothelial progenitors, some of which also underwent Cre recombination [91]. 

## 6. Regulation of Adult Muscle Homeostasis and Regeneration

Once muscle growth ceases, adult muscle is comprised of multinuclear, post-mitotic myofibers, and mononuclear quiescent muscle stem cells. In response to injuries or homeostatic needs, satellite cells become activated and support the formation of new myofibers, while a subpopulation will renew the quiescent pool for future needs. The skeletal muscle niche also includes a microvascular network, neurons, immune cells, and interstitial populations of mesenchymal progenitor cells, all of which participate in different stages of muscle maintenance and regeneration (for a comprehensive review see [99]). The role of BMP on adult muscle maintenance and repair has been investigated by analysis of homeostatic and regenerating muscle, respectively, by overexpression or deletion of different components of the pathway.

Two studies reported the effect of BMP administration on homeostatic (resting) muscle and both groups demonstrated that BMP promotes hypertrophy under these conditions (Figure 2). In the absence of injury, administration of Bmp7, Follistatin, or Bmpr1a led to increased Smad1/5 phosphorylation and subsequently fiber hypertrophy, mediated by the IGF and AKT/mTOR axes [95,97]. Hypertrophy was counteracted by simultaneous injection of Smad6 and Bmp7 or Smad6 and Follistatin [97]. Noggin administration or Smad1/5 knock-down also led to muscle wasting due to atrophy [95]. On the contrary, loss of the downstream Id1 and Id3 was not sufficient to induce a muscle phenotype [100]. Thus, in the absence of muscle injury (i.e., homeostatic muscle), exogenous BMP promotes muscle hypertrophy, raising the question of whether BMP is also involved in the generation of new muscle fibers following muscle injury and the associated muscle fiber loss and satellite cell activation.

Work from several groups established that quiescent satellite cells are pSmad-negative, but BMP signaling is active in adult regenerating muscle, in which satellite cells are activated in response to injury (Table A1) [44,45,76,91,100]. Of note, over a third of satellite cells co-express Smad1/5/8 and the Smad targets Id1 and Id3 at day 3 post-injury [100], while pSmad was also found in myonuclei and connective tissue cells [76]. BMP signaling through Acvr1, Bmpr1a, and Bmpr1b was identified as an important cue to stimulate cell cycle entry of differentiated myotubes and muscle fibers during newt limb regeneration [101]. Loss of Id1 and Id3 delays regeneration by reduction of satellite cell numbers and satellite cell proliferation at day 3 after injury [100]. Of note, Pax7 has been found to block premature differentiation of quiescent satellite cells by inducing the expression of Id2 and Id3, with quiescent satellite cells being Pax7+Id3+ [102]. Ono and colleagues used single fiber cultures to examine the involvement of BMP signaling in satellite cell activation. They first showed that Bmpr1a, pSmad1/5/8, Smad4, and noggin are all absent from the quiescent population and induced by culture, with noggin expressed in differentiating myogenin+ cells [45]. Furthermore, they found that exogenous BMP treatment inhibited differentiation of multiple cultures, and blockade of BMP signaling during cardiotoxin-induced muscle regeneration delayed the repair process [45]. Regeneration was also delayed when BMP was overactivated by relieving the miR-26a block on Smad1/Smad4/Id3 during muscle repair [98]. Similarly, hyperactivation of BMP signaling through the receptor Acvr1 led to a decrease in fiber size and an increase in fibrosis post-injury [103], showing that fine tuning of BMP levels is essential during the regeneration process (Figure 2). The effects of hyperactive Acvr1 were counteracted by rapamycin, which improved myofiber size and dissolved fibrosis [103]. Finally, BMP electroporation into regenerating rat muscles led to differentiation into osteogenic cells [104], corroborating in vitro results of BMP driving muscle cell lines into a bone fate [40,41,42,43,44,45,46,47,48,49,50]. 

Apart from satellite cells, BMP has been implicated in the regulation of additional cell populations during muscle regeneration. BMP inhibition in mesangioblasts contributed to enhanced grafting potential and muscle recovery [105]. Similarly, noggin treatment of bone marrow-derived mesenchymal stromal cells led to more dystrophin expression after engraftment into mice that lack dystrophin, a structural protein that maintains myofiber stability. On the contrary, Bmp4 treatment resulted in reduced dystrophin restoration [106]. Huang and colleagues found increased fat deposition in cardiotoxin-induced regenerating muscle of MyoD-Cre:Bmpr1a^f/f^ and Myf5-Cre:Bmpr1a^f/f^ mice. Their analysis suggested that BMP signaling in myo-endothelial progenitors repressed an adipogenic program both in developing and regenerating muscle [91]. 

Several studies also investigated the effects of conditional ablation of Smad4 on muscle regeneration (Table A1). Analysis of Myl1-Cre; Smad4^f/f^ mice showed that denervation injury significantly increased the level of atrophy in mutant mice, due to an increase in transcription of ubiquitin ligases, including Musa1 [95,97]. Later work demonstrated that inducible deletion of Smad4 with the use of a Pax7-CreERT2 allele led to significant impairment of muscle regeneration following cardiotoxin injection, with decreased fiber size and increased fibrosis [48]. On the contrary, knock-down of Smad4 in all muscle populations during regeneration (siRNA or miR-431 administration at D2 and D5 post-injury) leads to fiber hypertrophy [107]. However, these findings must be interpreted carefully, as the deletion of Smad4 abrogates TGF-β signaling, in addition to the BMP pathway. 

The role of BMP in aging or pathological muscles is less characterized. One study has shown that late middle-aged rats have lower BMP (measured by Bmp2, Bmp7, and pSmad1/5/8 levels) than young controls, while BMP levels can increase with exercise [108]. On the contrary, muscles exhibiting wasting associated with motor nerve transaction, degeneration or disuse were shown to have increased Smad1/5 phosphorylation and suppressed Smad6 and noggin, while administration of Bmp7 or Bmpr1a prior to denervation partially rescued the muscle wasting [97]. Deficiency in Fibrillin2, which is inversely correlated with BMP signaling, led to myopathic features in mice, including poor muscle architecture, reduction in muscle mass, and increase in centronucleated fibers, which are indicative of ongoing damage and regeneration [109]. The myopathic phenotype was exacerbated by Bmp7 overexpression but rescued by Bmp7 partial ablation or noggin overexpression [109]. In Duchenne muscular dystrophy (DMD), one of the most prevalent myopathies, Bmp4 was shown to be higher than in control samples and this increase was associated with the defective differentiation of myoblasts into myotubes [110]. Increased BMP was also observed in DMD iPSC-derived myoblasts [111]. Expectedly, noggin administration to an animal model of the disease induced the expression of myogenic markers (MyoD, myogenin) and mildly alleviated the disease phenotype [63]. Furthermore, Bmp4 was increased in patient myoblasts of the LMNA myopathy, while Bmp4 downregulation or Smad6 overexpression in mutant myoblast cultures promoted myogenic differentiation in vitro [112]. In Myhre syndrome, which is characterized by generalized muscular hypertrophy, there are Smad4 mutations associated with 8-fold increase in pSmad2/3 and 11-fold increase in pSmad1/5/8 [113,114]. Fibrodysplasia ossificans progressiva (FOP) is an extremely rare condition in which muscle and other soft tissues are spontaneously transformed into bone, and it is caused by a specific activating mutation in Acvr1 [115].

## 7. Perspectives

After several decades of work on BMP signaling in myogenic cells, it is clear that there is significant overlap in the function of this pathway during development and regeneration (Figure 2). In both scenarios, activation of the pathway seems crucial to maintaining expression of satellite cell master regulators like Pax3, while its timely repression is likewise required for differentiation and formation of muscle fibers. The main difference in BMP requirements between developmental and adult myogenesis seems to be that in the prenatal stage BMP has been shown to act as morphogen. However, several questions remain open regarding the levels and dynamics of BMP signaling in regeneration. Many groups have shown that a precise level of BMP signaling, tempered by the presence of inhibitory proteins, is crucial for initial stages of myogenic specification. We do not have the same detailed understanding of BMP levels in regenerating muscle, necessitating the development of new functional approaches, for instance that could modify the output of the BMP signaling pathway. It is also worthwhile to investigate the role of this pathway in other cell types that make up the skeletal muscle niche, including endothelial cells, immune cells, or mesenchymal progenitors, both during development and regeneration.

## Figures and Tables

**Figure 1 jdb-08-00004-f001:**
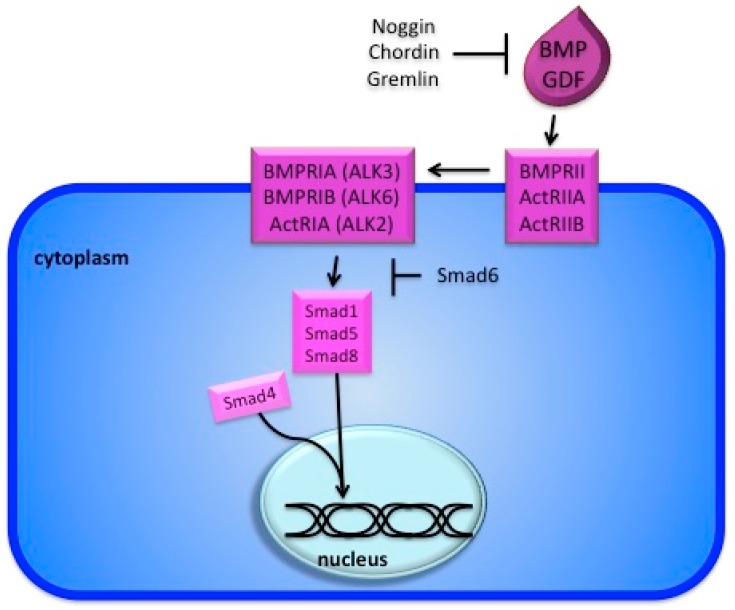
The BMP signaling pathway. Binding of the ligand to the type II receptor recruits the type I receptor and activates Smad 1/5/8 by phosphorylation. The later engages Smad4, which is shared with the TGFb pathway, and the complex translocates to the nucleus to regulate transcription of target genes (e.g. Id). The pathway is blocked by several inhibitors (e.g. Noggin) as well as the inhibitory Smad6.

**Figure 2 jdb-08-00004-f002:**
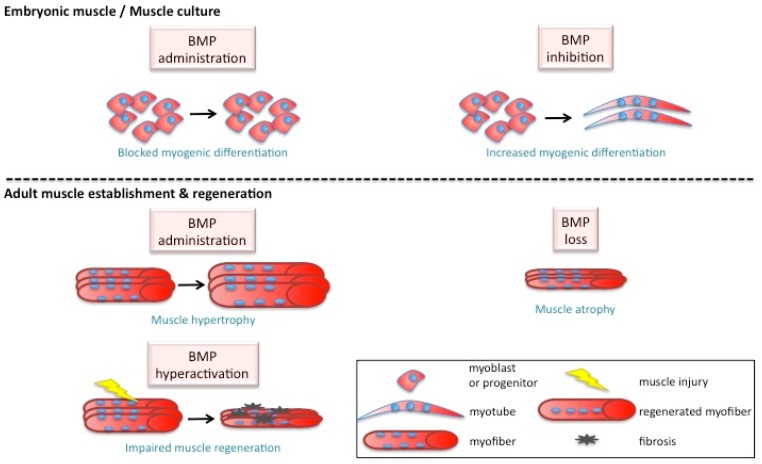
BMP effect on myogenesis. Fine tuning of BMP is essential for normal muscle growth. On one hand, BMP ensures muscle hypertrophy and its loss leads to muscle atrophy and wasting. On the other hand, hyperactive BMP leads to impaired regeneration and reduced fiber size post-injury. Furthermore, in culture and during development, inhibiting BMP is associated with increased myogenic differentiation, while excess Bmp blocks myogenic differentiation.

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
