# Peer review of "Bu-M-P-ing Iron: How BMP Signaling Regulates Muscle Growth and Regeneration"

_jdb, 2020, doi:10.3390/jdb8010004_

Round 1
Reviewer 1 Report
The review is extremely well written and thorough. My only suggestion, and not at all a demand, is that the authors prepare a table summarizing the muscle references and the salient points in those papers. Readership would be improved by reducing the amount of text that must be waded through
Reviewer 2 Report
This review by Borok et al aims to discuss the fundamental role of BMP signalling in muscle growth and regeneration. It is a very well written review and encompasses excellent detailed information on the role of BMP signalling in all aspects of muscle growth from fetal, prenatal and adult.
As the title mentions ‘regeneration’ the authors discuss the potential of BMP signalling in adult regeneration. The authors do mention the role of BMP signalling in diseases such as Duchenne Muscular Dystrophy and Myhre syndrome. However, it might be pertinent to mention something about Fibrodysplasia ossificans progressiva (FOP) and how a mutation of the ACVR-1 gene can have compounding effect of BMP on muscle cells to generate bone.
Minor comment,
Line 266-267 “In Myhre syndrome, which is characterized by generalized muscular hypertrophy, there are SMAD4 mutations associated with 8-fold increase in pSMAD2/3 268 and 11-fold increase in pSmad1/5/8 [111-112].” Try and standardise nomenclature throughout text ‘SMAD’ or ‘Smad’
Reviewer 3 Report
Borol et al. have written a nice review on BMP signaling in skeletal muscle. This is a difficult topic and the authors have presented evidence from a number of different models to condense what is known. I have a few suggestions that should be included for content and a few minor editorial comments.
Specific comments:
-A recent paper by Ferrazzo et al. (2019) in Muscle and Nerve should be mentioned. This paper addresses the effects of BMP-2 on primary muscle satellite cells (including proliferation, differentiation and osteogenesis). In addition, this group has a verified population of cells using PAX7 immunostaining.
-A brief section on BMP and osteogenesis/heterotopic ossification should be included. At the very least, the papers that used implanted BMP pellets (or similar) to determine the generation of bone within the skeletal muscle should be discussed. Some of these include Kusumoto et al. (2005) and Okubo et al. 2002.
-The authors should take care in the presentation of results related to immunocytochemistry. For example, in line 222, it is stated that “Noggin highly expressed in differentiating Myogenin+ cells [45]”. Reference 45 does not quantify Noggin expression. Immunocytochemistry can only be used for the detection of whether it is expressed or not.
Minor editorial:
-There are several paragraphs that would benefit from a summary statement and transition to the next paragraph.
-It is not expected that the authors cover BMP signaling in other tissues, but perhaps a statement on the other specific organ systems.
